# Predicting Critical Speed of Railway Tracks Using Artificial Intelligence Algorithms

Ana Ramos [1], Alexandre Castanheira-Pinto [1], Aires Colaço [1,*], Jesús Fernández-Ruiz [2] and Pedro Alves Costa [1]

1. Construct-Faculty of Engineering, University of Porto, 4200-465 Porto, Portugal; aramos@fe.up.pt (A.R.); up201108013@edu.fe.up.pt (A.C.-P.); pacosta@fe.up.pt (P.A.C.)
2. Department of Civil Engineering, University of La Coruña, Campus de Elviña, 15071 La Coruña, Spain; jesus.fernandez.ruiz@udc.es
* Correspondence: aires@fe.up.pt

**Abstract:** Motivated by concerns regarding safety and maintenance, the operational speed of a railway line must remain significantly below the critical speed associated with the track–ground system. Given the large number of track sections within a railway corridor that potentially need to be analyzed, the development of efficient predictive tools is of the utmost importance. Based on that, the problem can be analyzed in a few seconds instead of taking several hours of computational effort, as required by a numerical analysis. In this context, and for the first time, machine learning algorithms, namely artificial neural networks and support vector machine techniques, are applied to this particular issue. For its derivation, a reliable and robust dataset was developed by means of advanced numerical methodologies that were previously experimentally validated. The database is available as supplemental data and may be used by other researchers. Regarding the prediction process, the performance of both models was very satisfactory. From the results achieved, it is possible to conclude that the prediction tool is a novel and reliable approach for an almost instantaneous prediction of critical speed in a high number of track sections.

**Keywords:** artificial intelligence; railway dynamics; critical speed; numerical modeling; scoping tool

## 1. Introduction

According to the International Union of Railways (UIC), a total of 20 worldwide countries already have high-speed railways in operation in 2020, with a total length of 56.129 km, and 74.348 km more are under construction or planned. Thus, the expansion and improvement of the railway network, sometimes crossing soft soil formations, associated with the gradual increase in the operation speed, has brought new challenges in the design of railway tracks that in the recent past had only a theoretical meaning. One of the most significant aspects that deserves a deep analysis in the design phase is related to the critical speed phenomenon, once the maximum train speed should be limited by the propagation speed of surface waves through the track–embankment–ground.

When the train speed is close to the critical value, high amplifications of track displacements occur, having safety implications and a significant increase in track maintenance cost due to subgrade deterioration. The first studies regarding the dynamic amplification effects of the response due to a moving load on the surface of an elastic solid were conducted in the last century [1–7]. Promoted by Ledsgard's famous case study [8], the phenomenon described above has seen new developments in the latter years. From the theoretical point of view, the physical phenomenon concerned with the critical speed assessment is well established, existing several analytical and numerical approaches that can deal with the problem [8–12]. More recently, papers studying the efficiency of soil reinforcement techniques, commonly used in practical engineering, in the enhancement of critical speed have been published [13,14].

Despite the suitability of the presented numerical approaches in dealing with this phenomenon, the assessment of critical speed still remains very demanding from the computational point of view, especially when a large number of track sections need to be analyzed. Thus, viable alternatives should be formulated. In this sense, the introduction of artificial intelligence in critical speed prediction corresponds to a very innovative application that deserves great attention because it could become a powerful tool, with a very low computational cost, for predicting critical speed in railway tracks. In fact, the development of prediction tools based on artificial intelligence is a current topic, with accentuated growth in diverse fields of engineering. Focusing the exposition on railway problems, distinct applications of artificial intelligence techniques, essentially recurring to artificial neural networks (ANN), can be found, such as the work developed by Hanandeh, Ardah [15], Momeni, Yarivand [16] and Pham, Nguyen [17]. In more detail, the prediction of ground-borne vibrations induced by railway traffic involving ANN was addressed in the bibliography in recent years [18–21]. Jayawardana, Thambiratnam [22] and Hung and Ni [23] evaluated the effectiveness of trenches as a mitigation measure to deal with high levels of vibrations using ANN. A different application example can be found in Li, Meddah [24], where the relationship between track geometry to vehicle performance is also assessed through neural networks.

However, the number of works about the application of support vector machine (SVM) or regression is reduced when compared with ANN. Indeed, comparing both algorithms, there are very few works about SVM, even in the scope of civil engineering. One of the first was developed by Tinoco, Gomes Correia [25] in the scope of the implementation of the SVM applied to uniaxial compressive strength prediction of jet grouting columns. More recently, Huang and Kaewunruen [26] used support vector regression to study railway passenger comfort. On the other hand, Yuan, Zhu [27] used the concepts of the support vector regression to study the early detection of rail squats, and Balogun and Okine [28] used this algorithm to support their work in the detection of track geometry defects. Thus, despite few works in the railway field using the concept of SVM or even ANN, most of them are focused on track geometry defect detection and similar subjects. This work goes further, and this is the first step in the prediction of critical speed using simple machine learning algorithms. Thus, the application of ANN or SVM in this field is new and the machine learning algorithms allow to overcome the limitations of the "traditional methods" from the time of calculus until the parametrization. Therefore, this work proposes a new and innovative concept and approach to predict the critical speed of a railway track, which can be a valuable tool in the evaluation of the performance of the railway track.

In terms of paper organization, the paper begins by presenting an overview of the main concepts of artificial intelligence, focusing on artificial neural networks and support vector machine. After that, the phenomenon of critical speed is addressed through a conventional numerical approach, allowing the definition of a reliable dataset used in the derivation of the expedited prediction tool, as described in Section 4. It is worth noting that the numerical approach used in this paper is a 2.5D FEM-PML model, which has been experimentally validated by the authors in previous research works [29–31].

## 2. Expedite Prediction Tools Based on Artificial Intelligence: Generalities

### 2.1. Contextualization

Artificial intelligence (AI), machine learning (ML) or artificial neural network (ANN)/ support vector machine (SVM) algorithms are actual concepts, with strong growth over the last years. A simple scheme of the hierarchy relations is depicted in Figure 1. Artificial intelligence is assuming extreme relevance in the most diverse fields, and is expected to grow even stronger in the coming years. Generically, AI can be defined as a technique that enables machines to simulate human intelligence. As a subset of AI, machine learning is characterized by the use of statistical methods to enable machines to improve with experience, or, in other words, the system is trained to solve a given problem instead of explicitly programming the rules. Machine learning techniques can be divided into

two main groups: supervised and unsupervised learning. The main difference between both is that for the first one there is a dataset with inputs and the outputs are known, while unsupervised learning is a type of algorithm that learns from unlabeled datasets. For each type of learning, different categories of algorithms can be defined, as presented in Figure 1b. On a more restricted level, artificial neural networks and support vectors machine correspond to a subfield of supervised machine learning. However, ANN and SVM cannot be regarded as having equal importance and scope. In reality, ANN is a very broad area, involving a wide range of architectures, each designed for specific tasks and data types, while SVM is just a single algorithm.

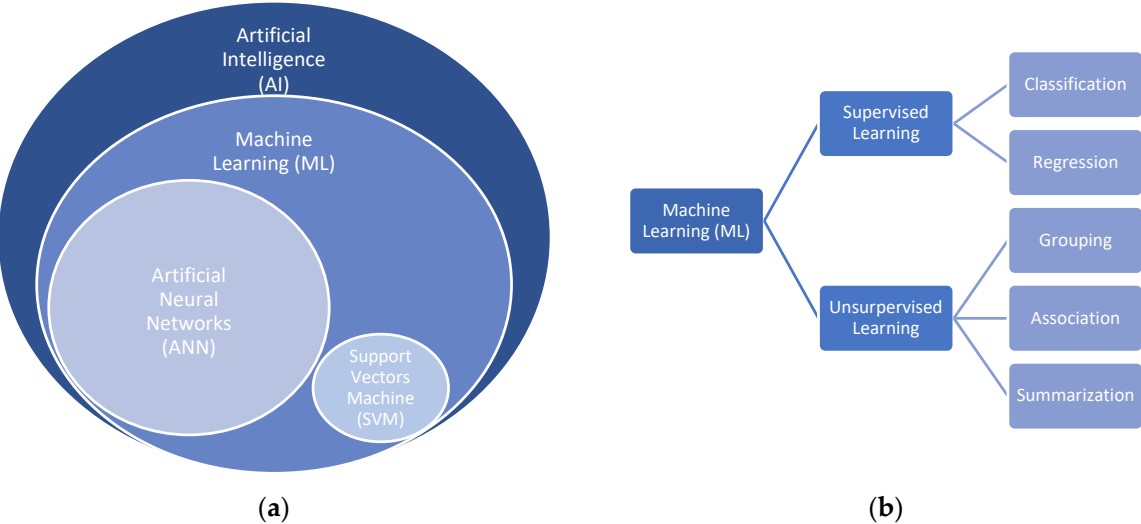

(**a**) (**b**)

**Figure 1.** Hierarchy relations: (**a**) relationship between AI, ML and SVM/ANN; (**b**) machine learning classification.

### 2.2. Artificial Neural Networks

ANN is one of the most widely used supervised machine learning techniques. These mathematical algorithms are inspired by the neural connections in the human brain. In terms of architecture, ANN is constituted by three different groups of layers: the first corresponds to an input layer and the last to an output layer. The intermediate group contains one or more hidden layers. The role of the first layer, the input layer, is to receive the information from the database, which is constituted by non-processing neurons. Contrarily, the hidden and output (which give the problem solution) layers are constituted by processing neurons, which are able to perform linear and nonlinear computations. Additionally, to the processing neurons, the hidden and output layers have additional nodes called bias. Typically, the layers are connected between them, with the output of a given layer serving as input for the subsequent layer. The interconnections established between neurons are defined as weight links. This characteristic architecture is usually denominated as a feedforward neural network; a general overview can be seen in Figure 2. In this analysis, the logistic activate function was selected to use in the prediction of critical speed.

In terms of model operation, a backpropagation algorithm is normally used as a training algorithm to learn from the datasets. ANN is an adaptative system that learns by using interconnected nodes (neurons) with variable weights. In more detail, the weights are assumed during the forward process, in which the information is being transferred from the input layer to the successive layers. This operation allows the calculation of the error between the target and the predicted value. In a subsequent step, the obtained error is propagated back in order to update the individual weights. Thus, the ANN uses a training algorithm to learn the database, modifying the neuron weights in the function of the error rate between the target and the predicted output. This process is repeated until the differences between the target and the predicted converge to an established level.

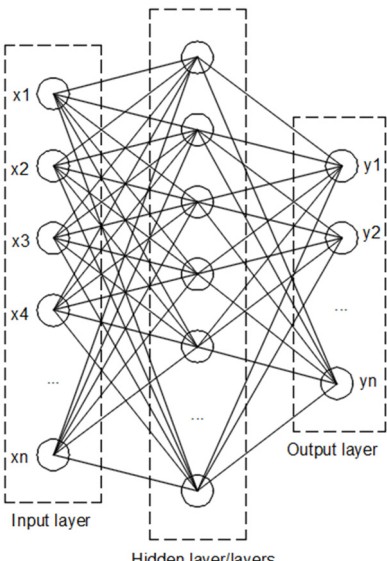

**Figure 2.** General overview of the ANN architecture (feedforward neural network).

Thus, ANN can learn from data to solve complex problems through the interpretation of nonlinear relationships between the variables of the problem. This means that the model is able to generalize and recognize patterns. Moreover, this algorithm can be applied and can predict a large amount of data accurately.

However, there are also some limitations. Neural networks are complex and, because of that, can require a significant amount of data, which can be time-consuming. Moreover, neural networks are also prone to overfitting. This means that it is necessary to validate the results with different datasets to identify the existence of this problem. The more common limitation is the difficulty in interpreting the results and understanding some decisions made by the algorithm. As a general comment, and in what concerns deep neural networks, a train test loss curve is a good way to identify overfitting issues.

*2.3. Support Vector Machine Algorithm*

The support vector machine (also known as SVM) is a supervised algorithm that can be used for classification and regression problems and even in outliers' detections. The SVM is able to solve nonlinear and linear problems. In a straightforward manner, the algorithm defines a line or a hyperplane that effectively segregates the data into distinct classes (in the case of a classification problem). Indeed, the SVM algorithm identifies the data points close to the line from both classes. These points are referred to as support vectors and the gap between them is known as the margin. The primary objective is to maximize this margin. The hyperplane of which the margin is maximum is called the optimal hyperplane (Figure 3).

This method offers notable advantages, including its effectiveness in high-dimension spaces and scenarios where the number of dimensions exceeds the number of samples. Additionally, this method is versatile since different kernel functions can be specified for the decision function. Nevertheless, this algorithm also has disadvantages, particularly when the number of features is superior (significantly) to the number of samples. Furthermore, the SVM does not directly provide probability estimative; indeed, these are calculated using an expensive five-fold cross-validation.

In this case, the work is developed as a regression problem and the method can be called support vector regression. Similar to classification problems, the model produced by the support vector regression depends only on a subset of the training data. This occurs because the cost function neglects samples whose predictions closely align with their target values. This algorithm can be considered very complex because of its high potential, and

its computing and storage requirements significantly increase with the number of training vectors (the core of an SVM is a quadratic programming problem (QP)).

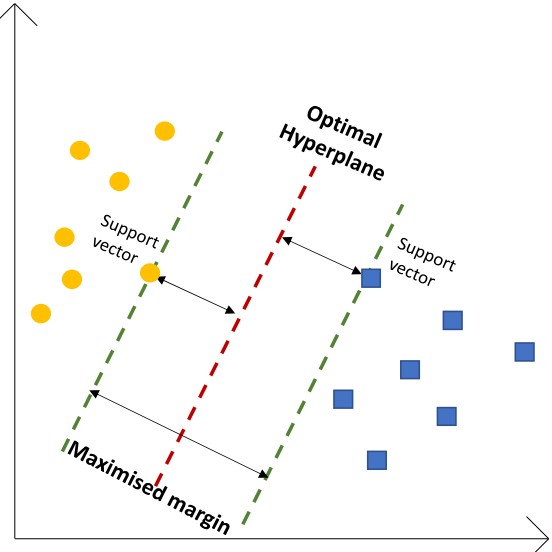

**Figure 3.** Definition of the SVM model.

The performance of the SVM can be increased through the tuning of the parameters. The more important parameters for SVM are called *C* and gamma. Parameter *C* (cost) controls the tradeoff between smooth decision boundary and classifying training points correctly. Indeed, a large value of *C* means that the user can obtain more training points correctly. The gamma parameter defines how far the influence of a single training example reaches. A high value of gamma implies that the decision boundary will be dependent upon the points that are very close to the line. This also means that some points that are far from the decision boundary are ignored. This occurs because closer points achieve more weight. On the other hand, if the value of gamma is low, this means that even far points can achieve considerable weight.

Hence, SVM holds a significant advantage in addressing nonlinear processes by incorporating a kernel function to project the original data into a high-dimensional linearly separable space. Therefore, SVM or SVR can be chosen as suitable options to build regression models when dealing with nonstationary variables. Moreover, when there is a clear margin, SVM works very well. This algorithm also exhibits greater effectiveness in high-dimensional spaces, particularly when the number of dimensions surpasses the number of samples. In terms of computational effort, SVM tends to be faster and more efficient when compared with ANN.

Nonetheless, it is important to note that SVM may not be well-suited for handling large datasets, and its performance tends to degrade when dealing with noisy data. Furthermore, its performance decreases when the number of features for each data point surpasses the number of training data samples.

When comparing both algorithms (ANN versus SVM), one of the primary distinctions lies in their approach to handling nonlinear data. SVM uses nonlinear mapping to make data linear separable, hence the kernel function is the key. ANN employs multi-layer connections and various activation functions to address nonlinear problems, such as this problem.

## 3. Database Definition

### 3.1. Numerical Assessment of Critical Speed

Critical speed is estimated by the assessment of the load speed that gives rise to the largest amplification of the track displacements. Therefore, a model needs to be run for different moving load speeds in order to establish a displacement amplification curve that is

then used to select the maximum value. Since analytical models and closed-form solutions are restricted in terms of geometry and model properties, a dynamic numerical approach is demanded to obtain the track displacements due to the traffic load.

In this sense, several distinct numerical approaches can be followed, being the finite elements method (FEM), the finite differences method (FDM), the method of fundamental solutions (MFS) and the boundary elements method (BEM) among the most popular. Regarding solution methods, both time domain analysis (explicit or implicit solution) and frequency domain analysis can be applied. It should be stressed that frequency domain analysis is confined to the solution of linear problems since it is based on the overlapping of effects.

On the other hand, transportation infrastructures, such as roads or railways, can be faced as infinitive and invariant structures. In such cases, the 3D wave propagation solution can be obtained through the combination of distinct plane waves that propagate along the structure development direction. Therefore, it is possible to apply a spatial Fourier transformation along that direction and to determine the 3D displacement field as a continuous integral of simpler bidimensional solutions, as

$$u^{3D} = \frac{1}{2\pi} \int\limits_{-\infty}^{+\infty} u^{2.5D}(k_1) e^{ik_1(x-x_0)} dk_1 \tag{1}$$

where $k_1$ is the longitudinal wavenumber and $x$ is the spatial coordinate.

This approach, usually called 2.5D, can be extended to different numerical techniques such as FEM, BEM or MFS, or to the combination between them [29–31]. The advantage of this method resides in the fact that only the cross-section needs to be discretized, without losing the 3D character of the problem. As a matter of fact, from the computational point of view, the method is quite efficient, since a small system of equations is solved several times (for different wavenumber/frequency) instead of solving an equation system with a large number of degrees of freedom (as is usual in 3D problems).

The introduction of the 2.5D concept into a FEM approach is a relatively straightforward process. In this process, the equilibrium of the system, written in terms of nodal variables, is given by (considering, in this case, the presence of external loads):

$$\left( \int\limits_z \int\limits_y B^T(-k_1) D B(k_1) \, dy \, dz - \omega^2 \int\limits_z \int\limits_y N^T \rho \, N \, dy dz \right) u_n(k_1, \omega) = F_n(k_1, \omega) \tag{2}$$

As highlighted in Equation (2), only the cross-section discretization is performed, since a Fourier transform is applied with respect to the longitudinal direction (x-direction). Therefore, the stiffness matrix, $[K]$, is computed by:

$$K^{FEM}(k_1) = \int\limits_z \int\limits_y N^T L^T(-k_1) \, D \, L(k_1) N \, dy \, dz \tag{3}$$

The advantage of employing the 2.5D approach resides in the ability to analytically compute the derivatives with respect to $k_1$. Therefore, matrix $[B]$ is derived from the product of the differential operator matrix $[L]$ (in the transformed domain) by matrix $[N]$, where:

$$L(k_1) = \begin{bmatrix} ik_1 & 0 & 0 & \frac{\partial}{\partial y} & 0 & \frac{\partial}{\partial z} \\ 0 & \frac{\partial}{\partial y} & 0 & ik_1 & \frac{\partial}{\partial z} & 0 \\ 0 & 0 & \frac{\partial}{\partial z} & 0 & \frac{\partial}{\partial y} & ik_1 \end{bmatrix}^T \tag{4}$$

Following the classical nomenclature of the finite element approach, the mass matrix is given by:

$$M = \int_x \int_y N^T \rho N dx dy \tag{5}$$

The drawback usually pointed out to the finite element approach when dealing with wave propagation problems is its inherent inability to work with unbounded domains, i.e., its requirement of truncation of the domain. This drawback can be correctly overcome by several approaches. In the present study, a PML (perfectly matched layer) approach is adopted. Thus, the combination of the 2.5D PML approach with the 2.5D FEM approach allows fulfilling Sommerfeld's condition, with a reasonable compromise between computational effort and accuracy. In more detail, the 2.5D PML is a layer that surrounds the interest domain (discretized with FEM) and allows absorbing the incident energy without reflection, as illustrated in Figure 4.

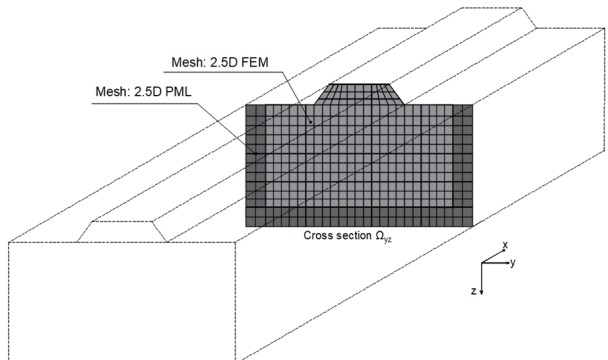

**Figure 4.** Infinite and invariant structure in one direction: 2.5D FEM-PML approach.

Since PML elements are similar to finite elements, the 2.5D solution is achieved by solving the following system of linear equations:

$$\left\{ [K_{FEM}(k_1)] + [K_{PML}(k_1, \omega)] - \omega^2 ([M_{FEM}] + [M_{PML}(k_1, \omega)]) \right\} u_n(k_1, \omega) = F_n(k_1, \omega) \tag{6}$$

where $K_{PML}$ and $M_{PML}$ are the PML stiffness and mass matrices, respectively. These matrices are obtained using the same formalism as the finite elements method but introducing complex stretching functions in order to avoid wave reflection. Details about the procedure, as well as its validation, can be found in the bibliography [29].

One of the advantages of the 2.5D approach resides in the simplicity of how the modeling can be extended to deal with moving loads, avoiding spurious transient effects that usually happen in 3D time domain analysis. In fact, extending the model to accommodate moving loads is a straightforward process, taking advantage of the shifting theorem of Fourier transformation [32]. Assuming that the load is given by the generalized Equation (7), which represents a load moving along the x-direction, with a speed $c$ and amplitude variable over time:

$$F(x, t) = F_x(x) F_t(t) \delta(x - ct) \tag{7}$$

Applying Fourier transformation with respect to the spatial coordinate $x$ ($k_1$ is the Fourier image of $x$), the previous equation takes the following form:

$$F(k_1, t) = F_x(k_1) F_t(t) e^{-ik_1 ct} \tag{8}$$

Considering now Fourier transformation with respect to time (with $\Omega$ the Fourier image of time variable $t$):

$$F(k_1, \omega) = F_x(k_1) F_t(\Omega - k_1 c) = F(k_1, \Omega - k_1 c) \tag{9}$$

which shows that all the equations already derived for the case of a stationary loading (the case with c = 0 m/s) can be transferred to the case of a moving load if $\omega$ can be transferred to:

$$\omega = \Omega - k_1 c \tag{10}$$

If the track is assumed as completely smooth, the invariance of the system prevents the development of the dynamic train–track interaction mechanism. In that case, the train can be replaced by a range of loads, respecting the train geometry, and moving at a constant speed *c*. In that case, Equation (10) is simplified to:

$$\omega = -k_1 c \tag{11}$$

*3.2. Database Scenarios*

As previously mentioned, the prediction accuracy of critical speed by means of an artificial intelligence algorithm strongly depends on the database used to train the algorithm. In this sense, the variables considered to develop the database were defined by taking into account the previous experience of the authors, which allowed to identify the most important parameters that conditioned the critical speed value [12,13]. Thus, in terms of track properties, and assuming that the rail adopted in most of the cases was the UIC60, the bending stiffness of the slab corresponded to the most influential parameter in the dispersion behavior of the track. Here, and assuming a relatively constant slab width and concrete properties (E), the slab's thickness was considered a key parameter to take into account.

The construction of a slab track in soft soil regions usually requires the consideration of an embankment to improve the conditions of the natural ground. Previous research studies performed by the authors [12] showed that embankment height plays a relevant role in terms of the dynamic behavior of the railway track when there is a contrast between the geodynamic properties of the embankment and the remaining domain. Given that, the thickness and stiffness of the embankment were also analyzed. Regarding the ground, simplified geotechnical profiles were considered, one being a homogeneous profile and another a dispersive soil. As easily perceived for these scenarios, both the propagating properties and the shallow soil-layer thickness constituted the principal parameters of the study. In order to clarify, and also summarize, all the geometrical and material properties assumed as variables in the study, an illustrative scheme can be seen in Figure 5.

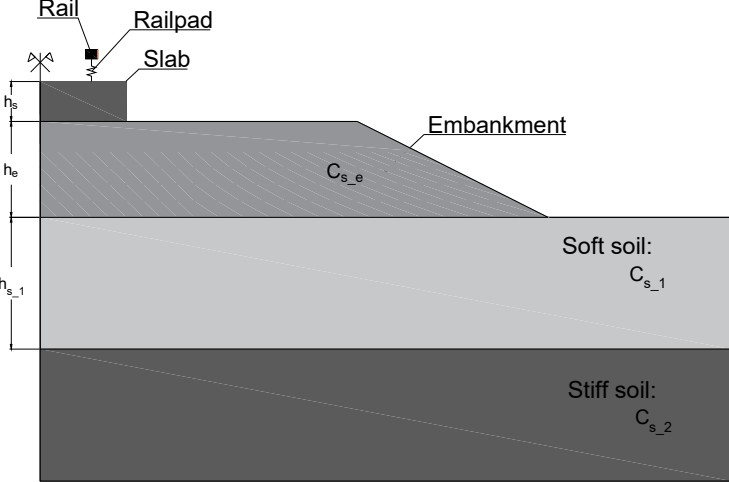

**Figure 5.** Illustrative scheme showing the material and geometrical properties assumed as variables.

Based on the previous considerations, Table 1 presents a detailed description of the track–ground properties adopted for database generation. The range of values assumed for each input parameter was chosen carefully to cover the most common situations found in engineering applications.

**Table 1.** Defined mechanical and geometrical properties of the track–embankment–ground system ($\rho$—mass density; $\xi$—damping factor; $\nu$—Poisson's ratio; E—elasticity modulus; Cs—shear wave velocity; h—thickness; bs—slab width).

| Rail | | UIC60 | | | |
|---|---|---|---|---|---|
| Railpad | K (kN/mm) | 50,000 | | | |
| | C (kNs/mm) | 22.5 | | | |
| Slab | E (GPa) | 30 | | | |
| | $h_s$ | (0.3, 0.4) | | | |
| | Width (m) | 3.2 | | | |
| | $\nu$ (-) | 0.2 | | | |
| | $\xi$ (-) | 0.01 | | | |
| | $\rho$ (kg/m$^3$) | 2400 | | | |
| Embankment | $h_e$ [m] | (1, 2) | | | |
| | $C_{s\_e}$ (MPa) | (169, 219) | | | |
| | $\nu$ (-) | 0.35 | | | |
| | $\xi$ (-) | 0.03 | | | |
| | $\rho$ (kg/m$^3$) | 1900 | | | |
| Natural ground | Homogeneous ground | | Layered ground | | |
| | | | 1st layer | | Infinite layer |
| | $C_{s\_1}$ (m/s) | (60, 90, 120, 150) | $h_{s\_1}$ (m) | (2, 6, 10, 14) | $C_{s\_2}$ (m/s)  (200, 350, 500, 1000) |
| | | | $C_{s\_1}$ (m/s) | (60, 90, 120, 150) | |
| | $\nu$ (-) | 0.35 | $\nu$ (-) | | 0.35 |
| | $\xi$ (-) | 0.03 | $\xi$ (-) | | 0.03 |
| | $\rho$ (kg/m$^3$) | 1900 | $\rho$ (kg/m$^3$) | | 1900 |

As previously mentioned, 6 parameters were adopted as variables ($h_s$, $h_e$, $C_{s\_e}$, $h_{s\_1}$, $C_{s\_1}$, $C_{s\_2}$), which, combining all possible scenarios, gave rise to 416 cases. The critical speed was computed for each one of them using 2.5D FEM-PML in order to establish the input database needed for the artificial intelligence algorithms.

## 4. Development of the Prediction Model

### 4.1. Initial Statistical Analysis

In order to understand the data and their patterns, a statistical analysis was performed. This pre-processing was essential to obtain good results and predictions. Thus, a univariate and multivariate analysis was performed. In the univariate analysis, the statistical measures (mean, mode, median, maximum, minimum, interquartile range, etc.) were obtained and analyzed. Moreover, in order to find patterns in the scope of multivariate analysis, the scatter plots were determined, along with correlograms (Figure 6). In the scatter plot, there was no evidence of a linear or monotonic relationship between the variables. Regarding the correlograms (Figure 6b), there was not indeed a strong relationship between the variables (to simplify, the variables were called $X_i$, as depicted in Table 2). The blank squares meant that the results did not show statistical significance (considering a *p*-value equal to 0.01). Furthermore, the results also showed that the variables were not correlated (the existing correlated variables showed very reduced values; e.g., ×2 with ×7 or ×2 with ×3), since it was not possible to find a pattern in the visualization of the data (Figure 6a).

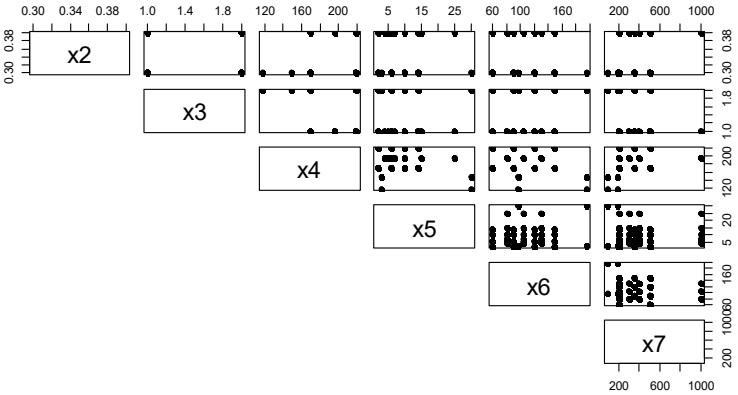

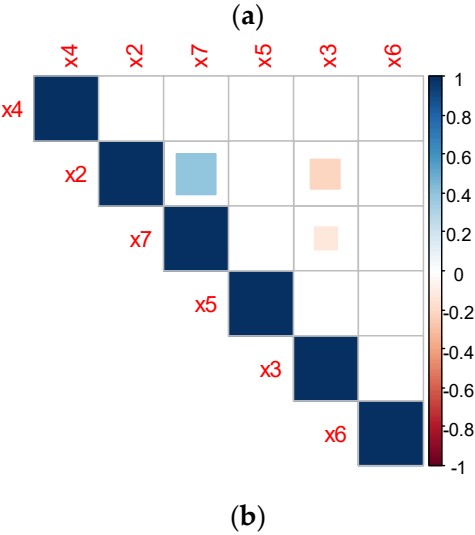

**Figure 6.** Multivariate analyses: (**a**) scatter plot; (**b**) correlogram.

**Table 2.** Identification of the variables.

| | | | |
|---|---|---|---|
| ×2 | Slab H (m) | ×6 | 1st layer Cs (m/s) |
| ×3 | Embankment H (m) | ×7 | 2nd layer Cs (m/s) |
| ×4 | Embankment Cs (m/s) | ×8 | Critical speed (m/s) |
| ×5 | 1st layer H (m) | | |

Thus, the variables described previously were kept.

### 4.2. Model Implementation

The dataset defined before was used to train a model that aimed to predict the correct outputs based on the inputs. The prediction model was composed of the training data and a learning algorithm (ANN or SVM) that would be used to emulate the physical system. The development of the intelligent prediction model involved different tasks, with different levels of complexity. Following, a brief resume of the most significant aspects is presented. The topics are numbered, allowing a clear interpretation of the development procedure of the prediction model.

#### 4.2.1. Input Variables Definition

The selection of the input variables was a common task of the different learning algorithms and their underlying engineering considerations in order to identify the most influential parameters in the critical speed of railway tracks. According to the previous

information, a total of six input variables were defined, involving properties of the track, embankment and ground:

- Track: slab thickness, $h_{slab}$ (m);
- Embankment: embankment thickness, $h_{embankment}$ (m) and stiffness (Young modulus), $E_{embankment}$ (MPa);
- Ground: first layer, shear wave velocity $C_{S,1}$ (m/s) and thickness $h_1$ (m); half-space: shear wave velocity $C_{S,inf}$ (m/s).

Regarding the neural networks, data normalization (or scaling) was an essential process, allowing to improve the backpropagation dynamics and, consequently, the optimization procedure. In this way, all the input variables, and correspondent output, were normalized in the interval between 0 and 1.

4.2.2. Parameters

The global behavior of ANN is determined by how its individual elements are connected and by the strength, or weights, assigned to those connections. These weights are automatically adjusted during the training process, following a specified learning rule until the neural network performs the desired task correctly. The learning process is a kind of spontaneous weight change. The number of neurons and hidden layers is essential for the performance of the neural network. Based on the dataset specificities, different combinations can lead to different results. Thus, it is essential to perform a sensibility study in order to identify an adequate combination. As the ANN model performance can only be evaluated after the training phase, the definition of the number of neurons and hidden layers corresponds to an iterative procedure.

Regarding SVM, there are two parameters that can influence the model and obtain results. These parameters can be adjusted and are usually called hyperparameters. *C* and gamma are two hyperparameters. In order to obtain a high accuracy of the model, it is essential to find the best values. They are very critical in building robust and accurate models. Indeed, striking the right balance between bias and variance is paramount to preventing the model from either overfitting or underfitting. Firstly, it is important to understand that the SCV establishes a decision boundary, which makes a distinction between two or more classes (Figure 3), in the case of a classification problem. The most critical aspect of SVM lies in determining the placement of the decision boudnary, especially when dealing with noisy data. In such scenarios, the decision boundary may need to be positioned very close to a particular class to accurately label all points within the training set. However, this proximity to the class can lead to reduced accuracy, as the decision boundary becomes highly sensitive to noise and minor fluctuations in the independent variables. Conversely, a decision boundary might be placed as far as possible for each class, costing some misclassified exceptions. This trade-off is effectively controlled by the parameter *C*. *C* can be defined as a hyperparameter in SVM to control errors. Thus, a low value of *C* means a low error. On the other hand, a large *C* means a large error. However, a low *C* or lower error does not mean a better decision boundary or a good model, that will depend upon the dataset's nature.

The gamma parameter is used with the Gaussian RBF (radial bases function) kernel. This kernel was selected to perform this analysis. In fact, in the case of linear or polynomial kernels, it is only necessary to use or define the *C* hyperparameter. In situations where the data points are not linearly separable, kernel functions are employed for transformation, and the gamma hyperparameter determines the curvature in a decision boundary. A high gamma value results in more curvature, while a low gamma value means less curvature. Once again, the choice between a low or high gamma value depends upon the dataset. Both parameters should be set before the training model. Therefore, it is important to identify the optimal combination of *C* and gamma. In cases where the value of gamma is large, the influence of *C* becomes negligible. Conversely, when gamma is small, *C* affects the model just as it affects a linear model. In this case, the best combination of *C* and gamma was selected.

### 4.2.3. Training

In this analysis, the holdout and the cross-validation approaches were used. The holdout approach was only used for the neural network and the cross-validation approach was used for the SVM and neural network. In the holdout approach, the training dataset was divided, considering a proportion into training ($p$) and test ($1 - p$). In this case, a $p = 70\%$ was adopted, which was close to the value usually recommended in the bibliography ($p = 2/3$). The cross-validation method usually shows very good results. In this approach, the observations were divided into $k$ subsets randomly with equal size. One of the folds was the holdout set (or the validation set) and the observations of the $k - 1$ partitions were used in the training process. This process was repeated $k$ times, using, in each cycle, a different partition to test. The final performance of the predictor was given by the mean of the observed performance over each subset of the test. However, the results did not significantly improve when the cross-validation method was used in the case of the neural network. In this case, k-fold cross validation was adopted. The parameter $k$ referred to the number of different subsets or folds into which given data were divided during the k-fold cross-validation. In this analysis, a value of $k = 10$ was selected.

### 4.2.4. Performance and Optimization

The last step of the intelligent prediction model implementation corresponded to the evaluation of its performance. For that, it was necessary that the quantification of the error associated with the training step. The following metrics were used: root mean squared error (RMSE), mean absolute error (MAE), mean square error (MSE), $R^2$ (coefficient of determination) and standard deviation (STD) between predicted and target values.

Based on that, and regarding the ANN model, an iterative study could be performed to identify the best combination of the number of neurons and hidden layers that led to the minimum error. From the studies developed, the combination hidden = (8, 3) was considered. This option meant that we had two layers with eight neurons plus three neurons. The output of the ANN is represented in Figure 7, where it is possible to detect the input variables, hidden layers and output variables.

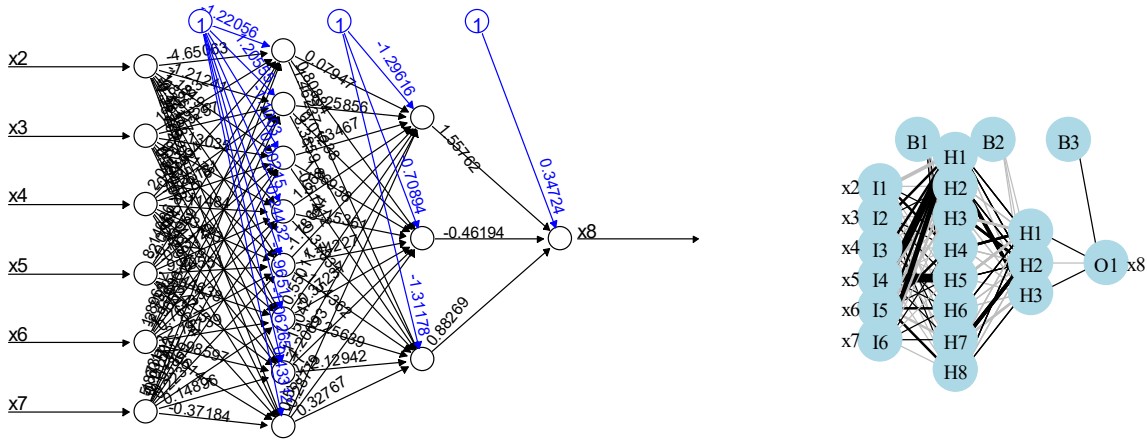

Error: 0.072958   Steps: 502

**Figure 7.** Neural network: architecture of the model developed.

The global behavior of the ANN model for the considerations above can be seen in Table 3, where a mark on the line drawn in black means a perfect correspondence between the predicted output and the corresponding target (observed values). The results are presented based on a scatter plot. The results show a good agreement between the predicted and observed values of the critical speed, reflected by the error metrics, such as R and RMSE.

To try to improve the results, the $k$ cross-validation method was implemented and used. Indeed, a good way to identify overfitting issues was to separate the dataset into three

subsets, including training, testing and validation. The training and testing set were used for the cross-validation process. The validation set was left untouched during the entire model development process. This validation set was used to validate and tune the model.

**Table 3.** Performance of the neural network.

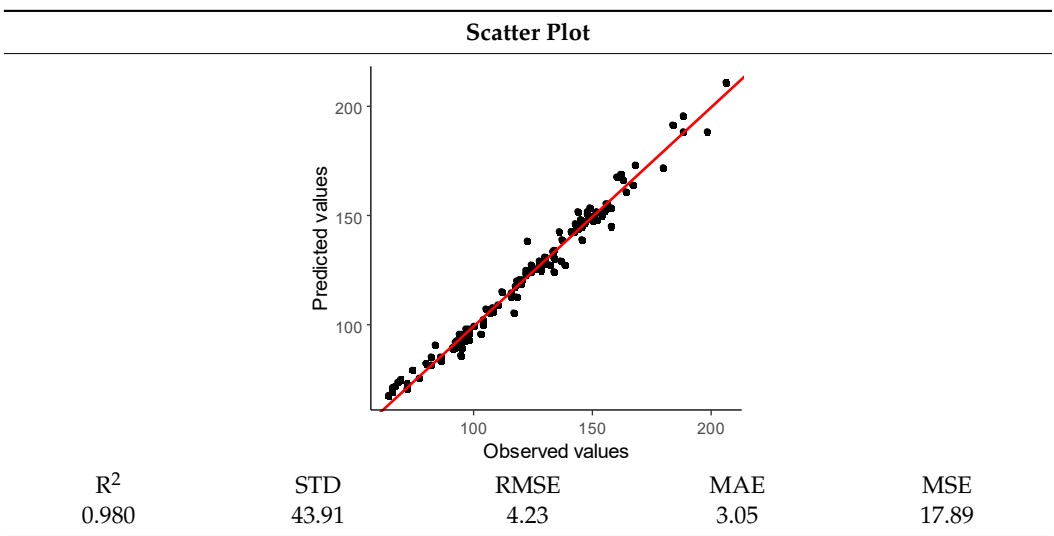

| R$^2$ | STD | RMSE | MAE | MSE |
|-------|-----|------|-----|-----|
| 0.980 | 43.91 | 4.23 | 3.05 | 17.89 |

As mentioned previously, in this analysis, a value of *k* = 10 was adopted. From the application of this approach, the results show that the RMSE slightly increased (RMSE = 4.55), as well as the MAE (3.23). This phenomenon may have been due to the significant dimension of the dataset, which had more than 400 inputs. The scatter plot of the observed results versus the predicted results of critical speed is depicted in Figure 8.

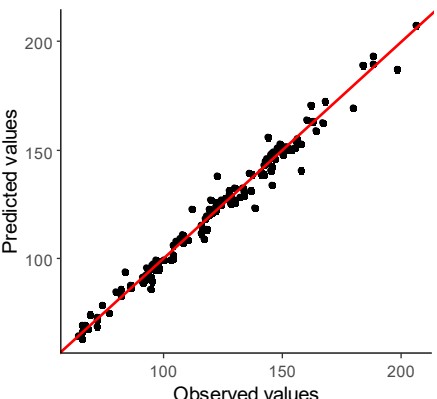

**Figure 8.** Neural network: scatter plot using the k cross-validation.

Based on the obtained results, it is important to mention that the model showed no indications of overfitting problems. In this case, the training data had a low error rate, as well as the test data. Moreover, cross-validation could be a powerful preventative measure against overfitting. Thus, the model showed high training accuracy and high validation accuracy. The validation results regarding the RMSE were: 3.79, 3.43, 5.31, 5.14, 4.59, 5.20, 4.37, 2.85, 3.73 and 4.77.

Despite the good results when applying the artificial neural network, the SVM algorithm was also tested and analyzed in detail. In what concerns the SVM, an iterative study was performed to identify the best combination of the *C* and gamma parameters that led to the minimum error. From the studies developed, the best combination corresponded to the pair of values (0.03, 16), where the first value corresponded to the *C* parameter and the

second one to the gamma parameter. The results of the hyperparametrization can be found in Figure 9.

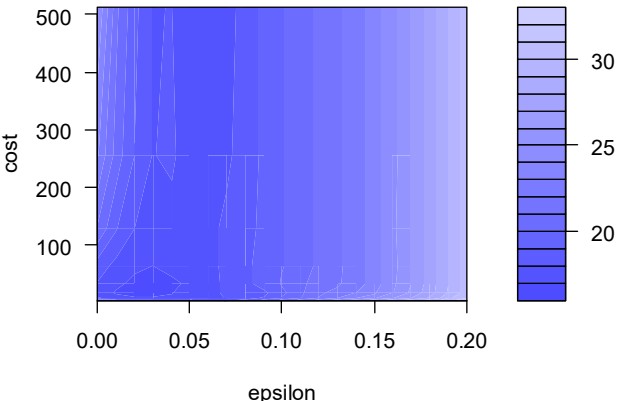

**Figure 9.** Performance of SVM: hyperparametrization.

Figure 9 shows the relationship between epsilon and cost. The value of epsilon determined the width of the tube around the estimated function (hyperplane): points that fell inside this tube were considered as correct predictions and were not penalized by the algorithm. The literature recommended an epsilon between $1 \times 10^{-3}$ and 1. The global behavior of the SVM model (considering the previous premises) can be seen in Table 4, where a mark on the line drawn in black meant a perfect correspondence between the predicted output and the corresponding target. Once again, the results are presented based on a scatter plot. The results show a very good agreement between the predicted and observed values, which were reflected by very low values of the error metrics. Indeed, for example, the RMSE of the SVM model was lower than the ANN model, which indicated the very good performance of this algorithm considering this dataset.

**Table 4.** Performance of the SVM model.

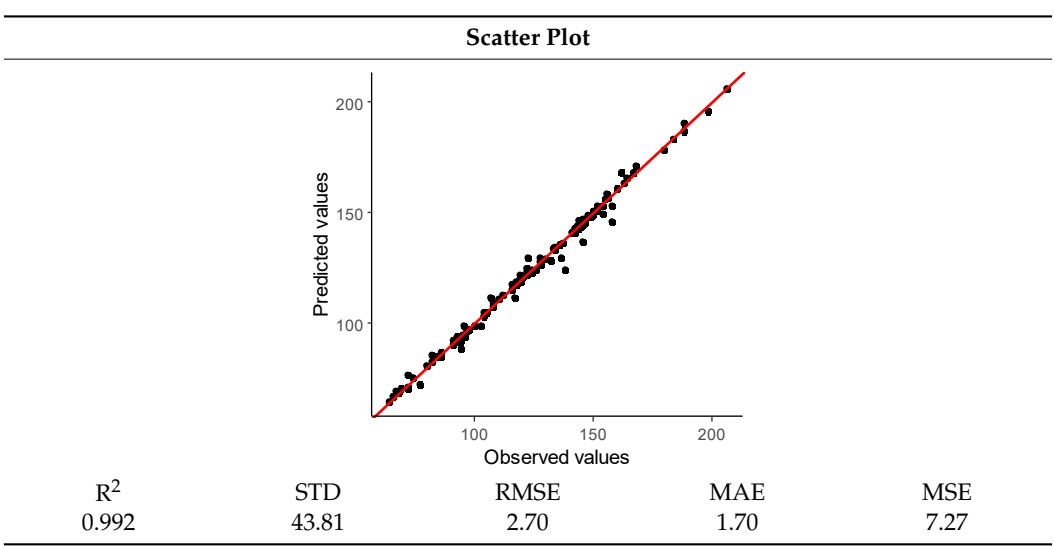

| $R^2$ | STD | RMSE | MAE | MSE |
|-------|-------|------|------|------|
| 0.992 | 43.81 | 2.70 | 1.70 | 7.27 |

Based on the previous results (metrics and scatter plots), it was possible to conclude that the model developed by the SVM algorithm led to better results. The model presented a very low value of RMSE and MAE and the scatter plot showed that the predicted values of the critical speed were very close to the observed values. It is important to mention that the high values of R did not mean an overfitting problem. The model showed a low error rate in the train, validation and test sets. This meant that the data variance was also low. Moreover, in SVM, to avoid overfitting, some data points were able to enter the

margin intentionally (soft margin) in order to prevent the overfitting problem. The gamma parameter also allowed to control this problem.

### 4.3. Validation of the Model: Application Examples

The preceding results demonstrate that the trained SVM model can accurately predict the critical speed of railway tracks based on the initially defined input variables ($h_s$, $h_e$, $C_{s\_e}$, $h_{s\_1}$, $C_{s\_1}$, $C_{s\_2}$). Therefore, this section is dedicated to validating the trained SVM model in different scenarios from those used in creating the previous database. Keeping this in mind and considering the aforementioned input variables, four new geotechnical scenarios were constructed, denoted as Case 1, Case 2, Case 3 and Case 4. Table 5 shows these new cases with values for each of the input parameters used to validate the effectiveness of the trained SVM algorithm.

**Table 5.** Variable properties for the new scenarios.

| Case | $h_s$ | $h_e$ | $E_{s\_e}$ | $h_{s\_1}$ | $C_{s\_1}$ | $C_{s\_2}$ |
|------|-------|-------|------------|------------|------------|------------|
| 1 | 0.4 | 1 | 150 | 4 | 80 | 200 |
| 2 | 0.4 | 2 | 150 | 10 | 130 | 300 |
| 3 | 0.3 | 2 | 200 | 5 | 50 | 350 |
| 4 | 0.3 | 2 | 200 | 5 | 170 | 350 |

Concerning the four new scenarios, the values adopted for the four parameters ($h_s$, $h_e$, $h_{s\_1}$ and $C_{s\_2}$) fell within the range of values considered in the database from the previous section. However, two parameters ($E_{s\_e}$ and $C_{s\_1}$) were outside this range, serving to evaluate the algorithm's ability to extrapolate. In this sense, as can be seen in Table 5, Cases 1 and 2 adopted input values within the range initially used to train the SVM algorithm, with a slight reduction in embankment stiffness to assess its impact on model performance. In Case 3, parameter $C_{s\_1}$ was below the minimum value used in the initial database, and in Case 4, $C_{s\_1}$ exceeded the maximum value from the initial database. The authors chose to vary parameter $C_{s\_1}$ in these last two cases due to its significant impact on critical speed.

Taking these considerations into account, the validation process involved comparing critical speeds computed through numerical simulations (with the model described in Section 3.1) and the SVM algorithm's predictions. Figure 10 displays the dynamic amplification factor (DAF) curve obtained from the numerical model (blue line) for all four analyzed cases, with the peak of the DAF corresponding to the critical speed. On these curves, the critical speed predicted by the SVM model is marked with a dashed vertical line. It is possible to observe in Figure 10, the high accuracy of the SVM algorithm in predicting the critical speed in all cases, even when the $C_{s\_1}$ is outside of the range of the training and testing database.

In addition to these four cases, a new database was defined in order to analyze in detail the performance of the model based on these new predictions. The new values for each one of the input parameters adopted to validate the SVM predicting algorithm were the following:

- $h_s$ = (0.4) m;
- $h_e$ = (1) m;
- $E_{s\_e}$ = (196) Mpa;
- $h_{s\_1}$ = (4, 5, 6, 7, 10, 15, 25) m;
- $C_{s\_1}$ = (80, 105, 130) m/s;
- $C_{s\_2}$ = (200, 300, 400, 1000) m/s.

By combining all the values into all the possibilities, 84 new geotechnical scenarios could be achieved.

The results were analyzed considering the scatter plot and metrics used previously ($R^2$, standard deviation, RMSE, MAE and MSE), as depicted in Table 6. The results showed an almost perfect alignment between the observed and predicted values. Indeed, these

very good results were reflected by the very low value of *RMSE* or *MAE*. This meant that this model was perfectly capable, in just a few seconds, to predict the critical speed of a slab track, with very high accuracy. These results also showed that the model was not trapped to local minima or having overfitting problems.

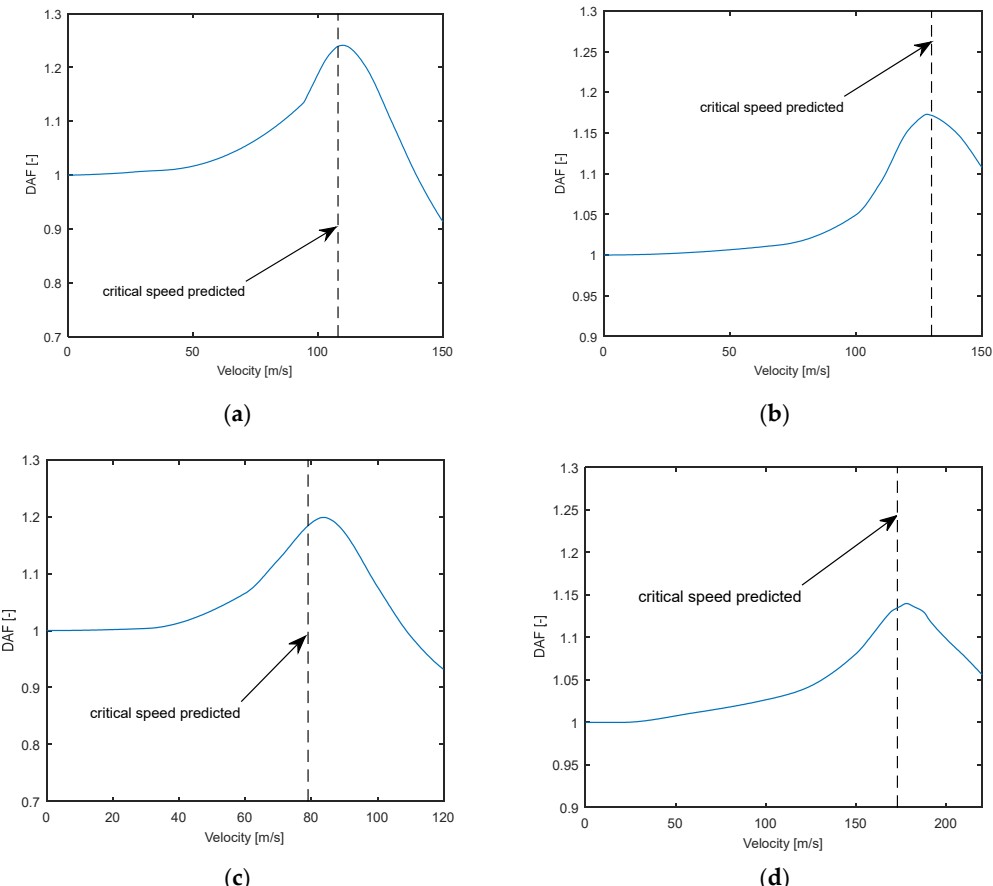

**Figure 10.** Dynamic amplification curve with the critical speed prediction of the SVM algorithm superimposed for: (**a**) Case 1 (critical speed prediction, 108.39 m/s); (**b**) Case 2 (critical speed prediction, 129.80 m/s); (**c**) Case 3 (critical speed prediction, 79.4 m/s); (**d**) Case 4 (critical speed prediction, 173.6 m/s).

**Table 6.** Performance of the SVM model.

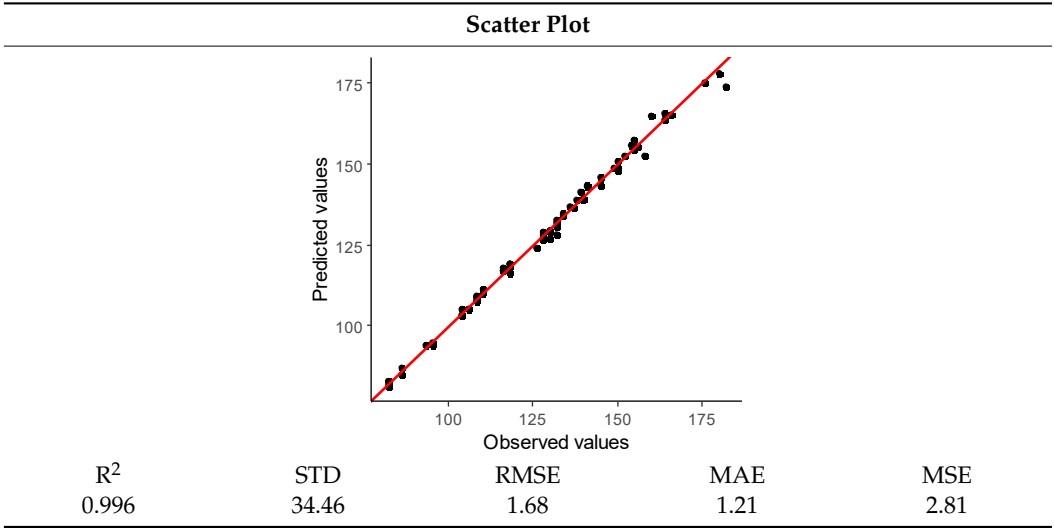

| R² | STD | RMSE | MAE | MSE |
|---|---|---|---|---|
| 0.996 | 34.46 | 1.68 | 1.21 | 2.81 |

*4.4. Sensitivity Studies*

To gain a deeper understanding of the model, a sensitivity analysis was conducted to assess which variables had a higher impact or influence on the model. In the case of ANN, it could be likened to standard regression models, where the weights connecting variables in the ANN play a similar role to the parameter coefficients in a regression model. These weights describe the relationship between the variables and determine their relative impact on the model's predictions.

In this context, the weights controlled how information was processed in the network. They served to suppress input variables that were less relevant in their correlation with the response variable. The opposite effect was observed for weights assigned to explanatory or important variables that had a strong positive relationship with the response variable. The determination of the relative importance of explanatory variables in a supervised neural network was based on the work developed by [23]. The fundamental idea was to identify all the weighted connections between the nodes of interest. The process involved identifying all the weights that connected the specific input node of interest as it passed through the hidden layer to the specific response variable. This process was repeated for all other explanatory variables until a list of all weights was found that were specific to each input variable. Indeed, the outcome of this analysis was a single value associated with each explanatory variable that described the relationship with the response variable in the model. Nevertheless, the algorithm only worked for neural networks with one hidden layer and one response variable. In this case, the method developed by Li et al. [24] was adopted. This method shares similarities with Garson's algorithm in that it relies on the connection weights between the layers of a neural network to access variable importance. However, Li et al. [24] offered an improvement over Garson's algorithm. It describes a connection weights algorithm and consistently demonstrates better performance in accurately representing the true variable importance, particularly in simulated datasets. The Olden method for calculating variable importance within neural network model involves a specific calculation. It computes the variable importance as the product of the raw input-hidden and hidden-output connection weights for each connection between an input and output neuron. These products are then summed across all hidden neurons. One notable advantage of this approach is that it preserves the relative contribution of each connection weight, taking into account both their magnitude and sign. In contrast, Garson's algorithm focuses solely on the absolute magnitude of connection weights. Given the developed model described in the previous sections, the application of this method led to the identification of the explanatory variables, and their importance is graphically represented in Figure 11. This graphical representation offers a clear visualization of the significance of each variable in influencing the model's predictions. The results show that the variable first layer H (m) (×5) was the variable with higher importance on the model, followed by the variable first layer Cs (m/s) (×6), second layer Cs (m/s) (×7), slab H (m) (×2), embankment H (m) (×3) and embankment $C_s$ (m/s] (×4). These results were expected since it was possible to state that it was the first layer that had a higher influence on the critical speed. Thus, these results were perfectly aligned with the expected physical behavior of the railway track in terms of critical speed [33].

Considering the model developed by the SVM algorithm, the impact of the variables on the model was also evaluated through the concept of the importance of the variables. Based on this concept, the variables were rated according to their importance, as depicted in Figure 12. The results show that the variables with higher importance were the first layer Cs (m/s) (×6) and first layer H (m) (×5), followed by the second layer Cs (m/s) (×7), slab H (m) (×2), embankment $C_s$ (m/s) (×4) and embankment H (m) (×3). Thus, it was possible to compare these results to the results from the ANN. Indeed, both models selected the same variables with higher importance. However, the importance of first layer H (m) was higher in the ANN model when compared with the SVM. This is, indeed, a major difference between the two models.

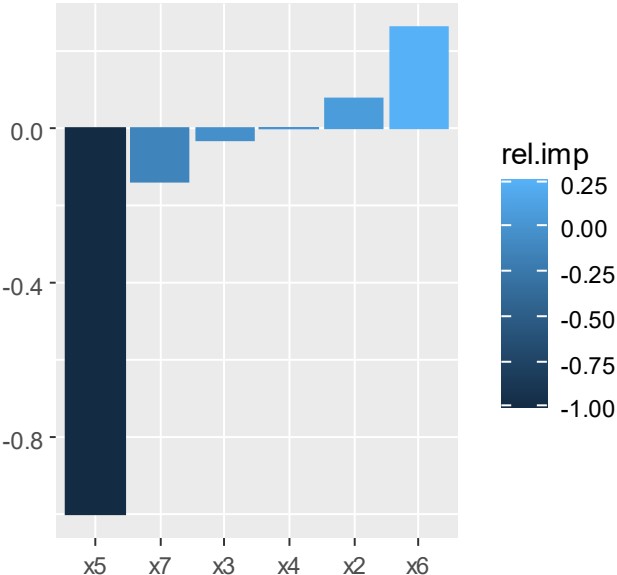

**Figure 11.** Importance of variables: neural network model.

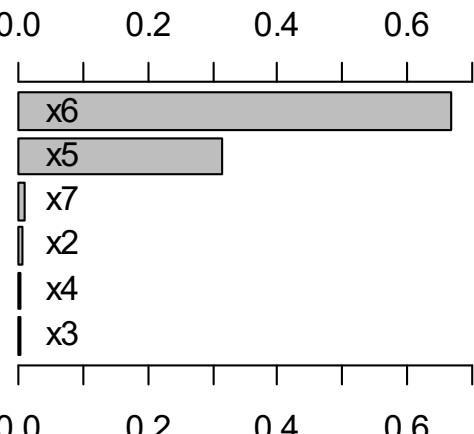

**Figure 12.** Importance of variables: SVM.

Moreover, the decision tree algorithm was also used to cross-validate the results. The decision tree allowed to quickly comprehend, by visualization tools, how the model was built, which included the variables with higher importance. This meant that this model and the results depicted in Figure 13 allowed to rank the variables with their importance. Although it was a different algorithm, these results may be compared with the previous ones since this algorithm, despite its simplicity, presented simple outputs and allowed to understand how the model was built through the selection of the variables. This was a significant advantage when compared with the ANN model, where the importance of variables was easy to obtain but it was not possible to see how the variables were selected. The obtained results are described in Figure 13. In this analysis a small value of *cp* (0.005) was selected in order not to hide possible important results. If a high value of *cp* was selected, the pruning would be significant, which would not allow observing the results so clearly. Based on the results depicted in Figure 13, it was possible to understand the variables ×5 (first layer H (m)) and ×6 (first layer Cs (m/s)) were extremely important in the building of the models to predict critical speed in a railway track.

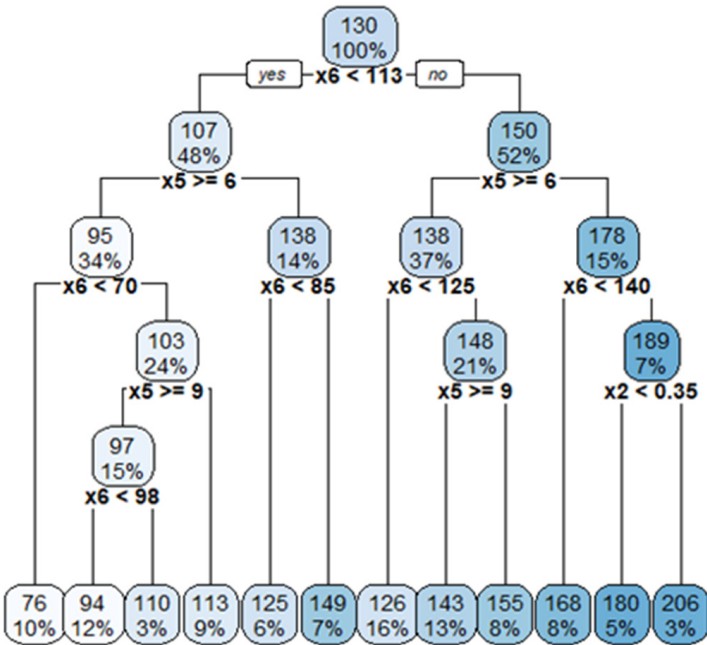

**Figure 13.** Decision tree model (*cp* = 0.005).

Considering the previous results regarding the importance of variables given by ANN and SVM algorithms, the following variables (with significant importance) were selected to build a new model: first layer H (m) ($\times$5), first layer Cs (m/s) ($\times$6) and second layer Cs (m/s) ($\times$7). The scatter plot and the metrics are presented in Table 7. The results showed that the performance of the model (using the SVM algorithm) decreased, since the *RMSE* and *MAE* were slightly superior when compared with the model with all variables included. Nevertheless, the $R^2$ value was still very high (0.979). From these results, it was possible to conclude that, despite the residual influence of some of the variables, they were important to improve the performance of the model and should not be excluded in a preliminary analysis.

**Table 7.** Performance of the model considering only three variables.

| Scatter Plot | RMSE | MAE |
| --- | --- | --- |
| *(scatter plot)* | 4.51 | 3.15 |

## 5. Conclusions

When planning new high-speed railway lines or enhancing existing ones for high-speed on soft soils, the phenomenon of critical speed becomes crucial. Computing the critical speed of a railway track requires high-standard numerical models, which involves huge computational strength and important technical specialization. To overcome these

limitations, artificial intelligence techniques, such as ANN (artificial neural network) and SVM (support vector machine), were used for first time in an innovative and novel way to predict the critical speed for dispersive media.

From a database with more than 400 cases and 7 important input variables, the performance of the developed models was evaluated through scatter plots and error metrics such as R, standard deviation, RMSE, MSE and MAE. From the results, it was possible to conclude that the SVM-based model was more accurate since the obtained error was very reduced. Indeed, the SVM and neural network algorithms usually showed very good results, and the selection of the best algorithm depended on the analyzed problem and type of data. In this case, the SVM algorithm, combined with the hyperparametrization., showed better performance.

Moreover, a sensitivity analysis was also carried out in order to check if the output of each model was aligned with the expected physical behavior. Thus, the variables were ranked according to their importance in the models. The results showed that the variables related to the first layer of the soil, such as stiffness and thickness, were the most important. However, the remaining variables should always be included in the dataset since their inclusion allowed to increase the accuracy of the model, despite its minor impact.

The approach presented in this paper introduces new methodologies and a very innovative, efficient and reliable tool to predict critical speed in just a few seconds, overcoming the problem of the high computational effort of numerical models. This efficiency is important in both academia and industry. Moreover, the new approach presented gives new insights into the relevance of geotechnical parameters on critical speed, which are important in regular engineering practice and can be used in the scope of the railway lines, considering different geometries and geotechnical scenarios.

Moreover, the data employed in this research (provided to any reader) serve as a comprehensive and valuable database for public administrations and railway engineers. They can be utilized as an initial reference for critical speed values during the early stages of design and planning, without the need to perform complex numerical models.

**Author Contributions:** Conceptualization, A.R., A.C.-P., A.C., J.F.-R. and P.A.C.; methodology, A.R., A.C.-P., A.C. and J.F.-R.; software, A.R., A.C.-P., A.C. and J.F.-R.; validation, A.R., A.C.-P., A.C. and J.F.-R.; formal analysis, A.R., A.C.-P., A.C. and J.F.-R.; investigation, A.R., A.C.-P., A.C. and J.F.-R.; resources, A.R., A.C.-P., A.C., J.F.-R.; writing—original draft preparation, A.R., A.C.-P., A.C. and J.F.-R.; writing—review and editing, A.R., A.C.-P., A.C., J.F.-R. and P.A.C. All authors have read and agreed to the published version of the manuscript.

**Funding:** This work was financially supported by: Base Funding (UIDB/04708/2020) and Programmatic Funding (UIDP/04708/2020) of the CONSTRUCT Instituto de I&D em Estruturas e Construções, funded by national funds through the FCT/MCTES (PIDDAC); Project PTDC/ECI-EGC/3352/2021, funded by national funds through FCT/MCTES; European Union's Horizon 2020 Programme Research and Innovation action under Grant Agreement No 101012456, In2Track3; Grant no. 2022.00898.CEECIND (Scientific Employment Stimulus, 5th Edition) provided by "FCT–Fundação para a Ciência e Tecnologia".

**Data Availability Statement:** The data that support the findings of this study are openly available at https://s.up.pt/0kbc.

**Acknowledgments:** The third author wishes to acknowledge the IACOBUS for the attribution of a scholarship for a 3 month research stay at U. Coruna. The fourth author also wishes to acknowledge the AUIP, as sponsoring institution of the Academic Mobility Scholarship Program, for the attribution of a scholarship for a 1 month research stay at University of Porto (Portugal).

**Conflicts of Interest:** The authors declare no conflict of interest.

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
