# Peer review of "Predicting Critical Speed of Railway Tracks Using Artificial Intelligence Algorithms"

_vibration, doi:10.3390/vibration6040053_

Round 1

Reviewer 1 Report

Figure 1a is confusing and could be controversial. Artificial Neural Networks is a very broad area which involves many different types of deep neural networks, such as MLP, CNN, RNN, PointNet, PointNet++, GNN, and DGCNN etc. Whereas SVM is just a single algorithm. It is very confusing when demonstrating them using same sized circles. Other than that, different people may prefer different hierarchy. As a result, if you want to keep Figure 1a, you will need a citation for it.

Figure 1b could also be controversial, since unsupervised learning is primarily used for tasks where you don't have labeled data and you want the algorithm to discover patterns or structure in the data on its own. Common techniques in unsupervised learning include clustering and dimensionality reduction. These techniques aren't typically used for regression tasks.

Based on Figure 2, it seems the neural network used in the study is a densely connected feedforward neural network (multilayer perceptron or MLP). It is suggested to change the name from ANN to either feedforward neural network or MLP to avoid confusion.

On line 135, the author states “Indeed, CNN, when compared to ANN, requires many more data inputs to achieve its novel high accuracy rate”. This statement needs a citation to support it.

If the reason to select ANN is because ANN requires less data for training, then why chose SVM which is not suitable for large dataset.

On Figure 6a, the tick marks for both axes are hard to read.

The dataset used in the study only has 400 data points but has 7 predictor variables. Why not use simpler models such as linear regression, or logistic regression etc. that are more immune to overfitting.

The author employs an R-value to evaluate the performance of the proposed models, and I'm wondering whether this R-value is equivalent to the R-squared (R²) value.

On line 427, the author states that “Based on the obtained results, it is important to mention that the model shows no indications of overfitting problems”. How did the author come to this conclusion.

On Figure 7a, does steps: 502 means 502 epochs?

For deep neural networks, a train test loss curve is a good way to identify overfitting issue. Please add it to your paper.

For machine learning models, a good way to identify overfitting issues is to separate your dataset into three subsets including training, testing and validation. The training and testing set are used for your cross-validation process. The validation set is left untouched during your entire model development process. This validation set is used to validate your finalized model when all hyperparameter tunning is done. Please add this to your paper.

For table 4, is the R (0.996) value the average of your 10-fold cross-validation? If yes please provide all the R values for your 10-fold cross-validation, and their standard deviation.

For table 4, what is the St.deviation? Is it the std for the R value?

Section 4.3 needs to be rewritten to make it clearer, since this section is critical to prove that the proposed method is not subject to overfitting.

 In table 5, what are the 4 cases? Are they 4 additional data points that you brought into your trained model for testing?

For Figure 10, a legend is needed. It is hard to tell what the blue curve represents. The dotted line represents your predicted value, then what is the actual value.

On line 505, what are the 84 new geotechnical scenarios? Are they 84 new data points untouched during the model development process, and the author is trying to use them as a separate validation set, like I mentioned in comment 11?

As shown in Figure 13, only predictor variable x6 and x5 shows significant importance. Then what is the point of including other variables? As I mentioned before, SVM has a training complexity of O(n^2) to O(n^3), and it may become intractable with large datasets and the model suffers from “curse of dimensionality”.

What is your R-value for table 7.

The conclusion section should be expanded. The author could discuss how the research could benefit both the academic and industry.

method description need to be improved especially section 4.3

Author Response

Dear reviewer,

please find our replies in the attached document.

We would like to express our gratitude for the relevant comments addressed in your report. These comments and suggestions helped us to improve the manuscript.

Thank you,

Aires Colaço

Reviewer 2 Report

Review V01 - comments

In all article for numbering the formulas I propose using simple brackets, e.g. (1), not [1];

L 97 – I propose to reformulate the sentences:  “The main difference between both is that for the first one there is a dataset with inputs and the out puts are known. Contrary, unsupervised learning is a type of algorithm that learns from unlabeled datasets.” into one sentence: “The main difference between both is that for the first one there is a dataset with inputs and the out puts are known, while unsupervised learning is a type of algorithm that learns from unlabeled datasets.”

L 164 - I propose to change: “its compute and storage requirements” into: “its computing and storage requirements”

L 182 – it is not clear: “In terms of calculation effort, the SVM is relatively memory efficient” – relatively more, or less, efficient?

L 402 – delete double word “metric metrics” in the sentence “The following metric metrics were…”

L 404 – why “standard deviation” is slanted and not normal?

L 415, L454 - it is not expressed what values are predicted and observed; similarly L 423  - observed results versus predicted results (results of what?)

L 518 -  correct the sentence: “…which variables that have a higher impact or influence on the model.” into: “which variables have a higher impact or influence on the model.”

L531 - correct the sentence: “This process is repeated for all other explanatory variables until is found a list of all weights” into: “This process is repeated for all other explanatory variables until a list of all weights is found”

L 596 – I suggest to use “requires” instead of “demands”

see the comments above

Author Response

(The authors gave the same response as above.)

Round 2

Reviewer 1 Report

The authors has addressed my previeous comments and it is now ready to be published. 

For your future research it is recommended to use more data points to test and develop your model. A dataset with only 400 data points but 7 variables and a complex model shows a good chance of overfitting. 

A simpler model like linear regression with lower R^2 may not indicate bad performance, instead it could indicate that the model is more robust when applied to external datasets.